# Moving forward through consensus: a national Delphi approach to determine the top research priorities in prostate cancer in Uganda

Andrew Sentoogo Ssemata [1,2] Richard Muhumuza [1] Janet Seeley [1,2] Dorothy Chilambe Lombe [3] Monde Mwamba [4] Susan Msadabwe [5] Amos Deogratius Mwaka [6,7] Ajay Aggarwal[8,9]

For numbered affiliations see end of article.

**Correspondence to**
Andrew Sentoogo Ssemata;
andrewssemata@yahoo.co.uk

## ABSTRACT

**Objective** To identify key areas for research in prostate cancer (PC) in the Ugandan context by establishing the major health system, socioeconomic and clinical barriers to seeking, reaching and receiving high-quality cancer care.

**Design** Modified Delphi Technique.

**Setting** Government and private-not-for-profit hospitals.

**Methods** We applied a two-stage modified Delphi technique to identify the consensus view across cancer experts. In round 1, experts received a questionnaire containing 21 statements drawn from a systematic review identifying the reason for the delay in accessing cancer care. Each statement was scored out of 20. Statements scoring ≥15 from over 70% of participants were prioritised for inclusion while statements for which <30% of participants gave a score of ≥15 were excluded. Sixteen statements were included in round 2 as they did not receive consensus for inclusion or exclusion.

**Results** We found that the top six research priority areas arise from challenges including: (1) lack of diagnostic services—ultrasound, laboratory tests and biopsy facilities; (2) high costs of services, for example, surgery, radiotherapy, hormone therapy are unaffordable to most patients, (3) lack of essential medicines, (4) limited radiotherapy capacity, (5) lack of awareness of cancer as a disease and low recognition of symptoms, (6) low healthcare literacy. The lack of critical surgical supplies, high diagnostic and treatment costs were ranked highest in order of importance in round 1. Round 2 also revealed lack of diagnostic services, unavailability of critical medicines, lack of radiotherapy options, high costs of treatments and lack of critical surgical supplies as the top priorities.

**Conclusion** These research priority areas ought to be addressed in future research to improve prompt PC diagnosis and care in Uganda. There is need to improve the supply of high-quality affordable anticancer medicines for PC patients so as to improve the survivorship from the cancer.

## STRENGTHS AND LIMITATIONS OF THIS STUDY

⇒ This is a novel Delphi study exploring research priorities for prostate cancer research in Uganda from various stakeholders in cancer care.
⇒ The study was informed by a recent systematic review, which provided insight into the statements explored in the Delphi technique.
⇒ Participants were recruited from various clinical, government-aided and private settings and geographical locations giving representativeness of the participants.
⇒ A limitation to this study was the low response rates in round 1 and round 2 as most stakeholders were not familiar with the Delphi technique, or how it works.

worldwide.[1] Prostate cancer (PC) is among the top three most common incident cancers in men (1.3 million incident cases) and the leading cause of cancer-related deaths for men in 56 countries including Uganda.[2]

A recent review has presented a worrying disparity in the mortality to incidence ratio from PC at 90% in Africa compared with 10% in North America.[3] For a curable cancer, these figures are stark. In the low and middle-income countries (LMICs), one of the challenges facing cancer control is the late diagnosis with advanced stage cancers at diagnosis.[4–6] The reasons that contribute to delayed diagnoses and advanced stage at presentation include delayed recognition of symptoms and signs of cancers by patients and primary healthcare professionals, inadequate health and cancer information for patients, poor continuity of care, lack of facilities for biopsy and histological diagnoses, costs and availability for treatments including drug therapies.[6–8]

One of the key challenges to delivering timely PC care is that the condition is largely

## INTRODUCTION

In 2020, there were 19.3 million new cancer cases and 10 million cancer deaths



asymptomatic in its early stages. African men also face much higher PC incidence, severity and greater risk of death than is seen in Caucasian populations, leading to much greater economic impacts on societies.[9]

From a cancer control perspective, to date, in Uganda, there has been much more emphasis on cancers affecting women, especially breast and cervical cancers than those affecting men such as PC.[10–12] Little attention has been given to the cancers affecting men especially PC despite increasing in incidence over the recent decades.[13–15] There is need to inform national control plans and develop the necessary infrastructure to manage this growing cancer burden. There is a need to prioritise key areas for research.

We present findings from the African Prostate Cancer— Disparities in Outcomes in Uganda study whose aim was to describe health system, socioeconomic and clinical factors that influence access to and outcomes of care for PC across hospitals in Uganda. The purpose of this Delphi process was to prioritise key areas for research in PC in the Ugandan context by establishing the major barriers to seeking cancer care, reaching cancer care and receiving high-quality cancer care. By defining future research priorities and investigations, this will help to support the future development of health system interventions to reduce inequalities and improve PC outcomes in hospitals across Uganda.

## METHODS

The Delphi technique is a relevant source of evidence in healthcare research foundational in the formation of consensus or the exploration of a field beyond existing knowledge.[16] It has been used in medicine to develop research priorities.[16–18]

A structured two-round modified Delphi technique with Ugandan clinical and health system research experts involved in PC care was conducted between December 2021 and July 2022. The Delphi technique was used to elicit consensus on the key research priorities in relation to the delays that may occur in receiving a diagnosis and/or treatment based on the Three Delays framework.[19 20] The 'Three Delays' Framework has been effective as a rapid health system assessment tool to understand complex multifactorial problems affecting access to care.[20] The Three Delays Framework focused on delays to: (1) seeking care—delays in recognising illness and deciding to seek appropriate medical help outside the home; (2) reaching care—delays in reaching an appropriate health facility and (3) receiving quality care—delays in receiving quality care after reaching the health facility.[21]

### Development of the survey

The Delphi survey was informed by two systematic reviews undertaken by the study team appraising primary quantitative,[15] and qualitative (Mwamba *et al*, under peer review) research studies that had sought to identify the factors influencing delays in cancer diagnosis and treatment as well as receiving quality care in Sub-Saharan Africa. From this literature review, 22 key factors contributing to delays were identified and represented priority areas for further investigation. These were translated into written statements that formed part of the Delphi consensus exercise for prioritising research in PC in Uganda (online supplemental appendix 1).

### Data collection

Data collection for round 1 of the Delphi process was conducted between December 2021 and February 2022 and round 2 of the Delphi process between March and July 2022. Participants were selected to ensure representation from different regions, healthcare sectors (primary and secondary); specialist disciplines (urology, oncology); academia and public health as well as patient representatives. Lists of participants to approach from both public and private health facilities in Kampala, Wakiso and Mukono districts were developed by the planning committee, which comprised two oncologists, and two health services research methodological experts who oversaw the design, execution and analysis of all phases of the study.

At present, there is only one major comprehensive cancer treatment centre in Uganda (the Uganda Cancer Institute, based at Mulago Hospital) with two active regional bases (in Mbarara, western Uganda, and Gulu; Northern Uganda),[6] and representation from each of these was sought. A snowballing approach was also undertaken where participants recommended additional stakeholders for participation. Agreement was reached regarding participant selection, consensus thresholds and survey format and question structure. For round 1, we approached participants from the Uganda Cancer Institute, government hospitals, Private-for-profit hospitals, Private-not-for-profit hospitals, Ministry of Health, and a university.

Due to low response rates (either non-response or incomplete questionnaires received from round 1) despite phone and face-to-face follow with the participants to ensure completion, it was necessary to invite an additional set of participants. It was decided by the research team to broaden the profile of participants to include cancer experts, clinicians, policy experts involved in cancer management from centres that provide cancer care services and work in liaison with the specialist centres included in round 1 to participate in round 2 of the Delphi process.

### Procedure

Experienced social science and clinical researchers conducted the data collection using a survey questionnaire. Participants were approached by a formal invitation to participate by email correspondence followed by a phone call and for some participants a face-to-face meeting. For those willing to participate, we collected demographic data, and a structured questionnaire was hand delivered to the participants. This was followed

up with a reminder phone call within a 4-week period to ensure completion of the question. Two sequential rounds of questionnaires were used for this Delphi process (online supplemental appendixs 1; 2).

Round 1 consisted of scoring the 22 statements using a 4-point scale. Each statement was assigned a score (out of 20) based on the four criteria, each scored out of 5, as below:

► *Feasible*—how easy is it to measure or assess this reason for the delay?
► *Large scale*—does this factor affect a considerable proportion of men with PC?
► *High Impact*—is this factor a significant cause of death or disability from PC?
► *Modifiable*—can this factor be reasonably addressed to improve the care of men with PC?

Once all the questionnaires for round 1 were received, we compiled the rankings for each statement. Based on the criteria by Schneider *et al*,[22] statements were selected as follows.

Statements where 70% of participants gave a score of ≥15 were included as research priorities. Statements for which <30% of participants gave a score of ≥15 were excluded at this stage.[22] The rest of the statements went forward to the second round of the Delphi consensus process. The information provided in the round 1 of the Delphi technique was collated and summarised in REDCap software,[23] to enable formal analysis, and to formulate a second questionnaire with fewer statements based on the selection criteria above.

Round 2 of the Delphi included 18 statements. An electronic questionnaire designed in REDCap was generated and emailed to 42 participants. Participants were followed up with a reminder phone call within a 4-week period and through face-to-face meetings by the study team. Recurrent emails and phone calls were made to those who had been contacted and had not completed the survey. Each participant was provided a time compensation of 30 000 Ugandan shillings at the end of each completed questionnaire.

### Patient and public involvement

Patients were not involved in the design, or conduct, or reporting or dissemination plans of this research.

### RESULTS

Ten and 12 respondents completed the survey questionnaires in round 1 and 2, respectively. In round 1, a total of 30 clinicians were invited through email and face to face meetings, of which 10 (34%) participated and completed the survey. The majority of the participants in this round were oncologists n=4 (40%) and cancer researchers n=3 (30%). Most of the participants worked in a cancer care facility. In round 2, a total of 42 clinicians were invited through email, online and face to face meetings, of which 12 (29%) participated and completed the survey. Most of the participants in this round were radiologists n=3

(25%) and general practitioners n=3 (25%). Most of the participants in round 2 operated from a general healthcare setting. Of the 22 people who participated in rounds 1 and 2, three participated in both rounds and seven participants participated in round 1 only. The characteristics of the respondents in rounds 1 and 2 are presented in table 1.

Following the first round of the Delphi technique, we excluded three statements of research priorities from the second-round questionnaire where <30% of participants had given them a score less than 15. These statements were related to barriers such as staff motivation, inadequate training, etc. Two of the barriers scored greater than 70% and were to be included within our final selection of research priorities. These were *lack of awareness of cancer as a disease* and *poor healthcare literacy* (table 2).

In round 2, of the 16 statements evaluated, four statements reached the consensus target (>70% of respondents scoring the statements ≥15) as shown in table 3. These were (1) lack of diagnostic services (X-ray, ultrasound, labs (eg, prostate-specific antigen testing, biopsy facilities), (2) lack of availability of critical medicines, (3) lack of radiotherapy options (brachytherapy/teletherapy) and (4) cost of treatment, for example, surgery, radiotherapy, hormone therapy. In total, six research priorities were identified during the two-round Delphi technique.

### DISCUSSION

The purpose of this Delphi technique was to prioritise key areas for research in PC in the Ugandan context and will form the basis of recommendation for cancer control planning and research investments. The Delphi process sought to prioritise those factors that had the greatest impact on achieving timely access to high-quality care could be investigated empirically and for which interventions could be developed and implemented.

Our study invited cancer experts, clinicians and policy experts in and outside cancer management and care to evaluate the top research priorities for PC care in Uganda. The difference in the panels between rounds 1 and 2 was favourable as including participants from different healthcare systems, professional backgrounds and specialities created a relatively diverse panel and suitable representation that provided insightful ideas and perspectives towards the priority research needs for PC in the Ugandan context. There could be potential bias introduced by two different panels participating in the two Delphi rounds as it is often impractical to secure the participation of all key knowledgeable individuals for all the Delphi rounds and consensus meeting. However, we were able to include participants from different health service strata (primary and secondary care), professional backgrounds and specialities to provided insightful perspectives towards prioritising research needs for PC in the Ugandan context. The priorities chosen reflect important challenges, including cancer awareness, increasing diagnostic and treatment capacity as well as

**Table 1** Showing characteristics of participants in round 1 and 2 of the Delphi process

| | **Round 1** | | |
|---|---|---|---|
| | **Gender** | **Profession** | **Organisation** |
| 1 | Male | Oncologist | Cancer Institute, Kampala |
| 2 | Male | Oncologist | Cancer Institute, Kampala |
| 3 | Male | Oncologist | Cancer Institute, Kampala |
| 4 | Male | Cancer researcher and administrator | Cancer Institute, Kampala |
| 5 | Male | Oncologist | Cancer Institute, Kampala |
| 6 | Male | Public health specialist and policy expert | University/Ministry of Health, Kampala |
| 7 | Male | Medical doctor/general practitioner | Government Hospital, Kampala |
| 8 | Female | Researcher and clinical officer | Government Hospital, Kampala |
| 9 | Male | Urologist | Cancer Institute, Kampala |
| 10 | Female | Surgeon and cancer researcher | Government Hospital, Kampala |
| | **Round 2** | | |
| 1 | Female | Urologist | Private for-profit hospital, Kampala |
| 2 | Female | Surgeon and cancer researcher | Government Hospital, Kampala |
| 3 | Female | Researcher and clinical officer | Government Hospital, Kampala |
| 4 | Male | Medical doctor/general practitioner | Government Hospital, Kampala |
| 5 | Male | Urologist | Government Hospital, Kampala |
| 6 | Male | Medical doctor/general practitioner | Private not-for-profit hospital, Kampala |
| 7 | Female | Radiologist | Private not-for-profit hospital, Kampala |
| 8 | Female | Radiologist | Private not-for-profit hospital, Kampala |
| 9 | Male | Surgeon | Government Hospital, Mukono |
| 10 | Male | Medical doctor/general practitioner | Government Hospital, Mukono |
| 11 | Male | Surgeon | Private for-profit hospital, Kampala |
| 12 | Male | Radiologist | Private not-for-profit hospital, Kampala |

addressing the high costs of care, which are not likely to have substantially changed with additional Delphi rounds or participants.

Six key domains were identified: (1) lack of awareness of cancer as a disease and recognition of symptoms, (2) poor healthcare literacy, (3) lack of diagnostic services, (4) lack of availability of essential medicines, (5) lack of radiotherapy facilities and (6) cost of treatment.

With respect to these six areas, there remains a paucity of literature to date in the Ugandan context but are important issues as outlined below.

### Lack of awareness of cancer as a disease and poor recognition of cancer symptoms

Poor knowledge and misconceptions regarding PC and screening among Ugandan men[24] is likely to contribute to diagnosis with advanced clinical stages (stage III and IV) of PC.[25] A lack of awareness about cancer and low health literacy has been reported as one of the major reasons for delays in seeking quality cancer care in sub-Saharan Africa.[15] Similarly, the lack of awareness of PC, its symptoms and its consequences definitely affects the uptake of screening and seeking specialist help that provides an opportunity for early detection of PC. The lack of adequate information and counselling on PC has

been associated with the considerably low uptake of PC screening among Ugandan men.[26] Developing appropriate interventions for improving awareness and knowledge about PC are critical in supporting early detection and treatment outcomes of men who develop PC.

### Low health literacy (when, how and where to seek services)

Healthcare literacy is defined as the degree to which individuals have the capacity to obtain, process and understand basic health information and services needed to make appropriate health decisions and to successfully navigate the healthcare system.[27 28] Poor healthcare literacy limits access to care, interaction with health service providers and illness management as people with low healthcare literacy are more likely to have poorer use of health services and, therefore, experience poorer health outcomes.[29] A recent study among PC survivors in Uganda has shown that inconsistent information or complete lack of information are common experiences of the PC survivors can affect their health outcomes.[30 31] Patients with limited health literacy may have limited knowledge and understanding of health and when coupled with lack of established follow-up mechanisms and inadequate social care support services experienced by many cancer

**Table 2** Ranking of round 1 questionnaire statements

| | Statement | Ranking (% of participants scoring reason≥15) |
|---|---|---|
| 1 | Lack of awareness of cancer as a disease and recognition of symptoms | 90 |
| 2 | Poor healthcare literacy (when, how and where to seek services) | 90 |
| 3 | Lack of critical surgical supplies | 70 |
| 4 | Cost of diagnostic investigations | 70 |
| 5 | Cost of treatments for example, surgery, radiotherapy, hormone therapy | 70 |
| 6 | Lack of workforce (basic numbers low and inadequate training of staff) | 70 |
| 7 | Accessibility of care (long distance/travel times to access specialist services) | 60 |
| 8 | Difficulties with healthcare coordination between regions and hospitals as patients referred for specialist investigation and treatment | 60 |
| 9 | Lack of availability of essential medicines | 60 |
| 10 | Cost of accessing healthcare (eg, cost of accommodation and transport needed to receive treatment from centralised services) | 60 |
| 11 | Misdiagnosis of cancer at lower system levels (eg, primary care, district hospital) | 50 |
| 12 | Lack of diagnostic services (X-ray, ultrasound, labs (eg, prostate-specific antigen testing, biopsy facilities) | 50 |
| 13 | Lack of radiotherapy options (brachytherapy/teletherapy) | 50 |
| 14 | Preference for traditional, complementary, and complementary medicine | 40 |
| 15 | Lack of trust in healthcare and patients' citizens' rights (perceived quality; attitudes of healthcare workers; previous bad experience, eg, patients being turned away or refusal to refer; adequate consent) | 40 |
| 16 | Personal and professional obligations (financial and social implications to the patient and their families of seeking care and undergoing treatment) | 40 |
| 17 | Lack of social capital to support cancer journey especially where patients must travel for care (relationships, support from family, friends, colleagues) | 40 |
| 18 | Patient fitness and treatment toxicity | 40 |
| 19 | Stigma associated with cancer diagnosis or severe illness/fears and beliefs around cancer | 30 |
| 20 | Communication/language barrier between healthcare staff and patients | 20 |
| 21 | Staff motivation and burnout | 20 |

patients in Uganda, this reduces their autonomy in self-care and decision-making.[6]

### Lack of diagnostic services
A lack of basic imaging and pathological services results in delays in confirming a cancer diagnosis and completing staging to support management options. Such delays have a knock-on effect on the timely delivery of treatment leads to high mortality, morbidity and low quality of life.[25 32 33] Lack of diagnostic services has led to delayed cancer diagnosis being a common in the Ugandan setting.[31] Additionally, where the service has been obtained, the high costs of diagnostic investigations compounded by the poor social-economic status of the patients has resulted in enormous delays in seeking PC care among Ugandan men.[6]

### Lack of availability of essential medicines
A recent review has reported major barriers in access to core cancer medicines worldwide, with high prices are

major barrier including medicine included in the WHO essential medicines list.[34] Even relatively low-cost medicines (compared with chemotherapy and molecularly targeted anti-cancer agents) such as Goserellin are largely unavailable to patients in the Ugandan setting due to recurrent stock-outs and majority being costly.[6] A recent recommendation has suggested increased repurposing of existing drugs such as Metformin, Valproic acid initially intended for other conditions to treat PC.[35]

### Lack of Radiotherapy facilities
This is a core treatment for PC particularly in high-risk disease. However, radiotherapy is not commonly available in sub-Saharan Africa with limited experience for radiotherapy and brachytherapy, insufficient infrastructure as well as limited trained personnel and training opportunities.[36] Its low provision in Uganda has made the provision of treatment unachievable for many cancer sufferers. It has been previously noted that Uganda needs more than

**Table 3** Round 2 ranking

| Rank | Barrier | Ranking (% of participants scoring reason≥15) |
|---|---|---|
| 1 | Lack of diagnostic services (X-ray, Ultrasound, labs (eg, prostate-specific antigen testing, biopsy facilities) | 92 |
| 2 | Lack of availability of critical medicines | 92 |
| 3 | Lack of radiotherapy options (brachytherapy/teletherapy) | 83 |
| 4 | Cost of treatment for example, surgery, radiotherapy, hormone therapy | 83 |
| 5 | Lack of critical surgical supplies | 67 |
| 6 | Cost of accessing healthcare (eg, cost of accommodation and transport needed to receive treatment from centralised services) | 67 |
| 7 | Accessibility of care (long distance/travel times to access specialist services) | 58 |
| 8 | Cost of diagnostic investigations | 58 |
| 9 | Misdiagnosis of cancer at lower system levels (eg, primary care, district hospital) | 50 |
| 10 | Patient fitness and treatment toxicity | 50 |
| 11 | Difficulties with healthcare coordination between regions and hospitals as patients referred for specialist investigation and treatment | 50 |
| 12 | Lack of trust in healthcare system and patients' citizens' rights (perceived quality; attitudes of healthcare workers; previous bad experience, eg, patients being turned away or refusal to refer; adequate consent) | 33 |
| 13 | Lack of workforce (basic numbers low) | 33 |
| 14 | Preference for traditional, complementary and alternative medicines | 25 |
| 15 | Personal and professional obligations (financial and social implications to the patient and their families of seeking care and undergoing treatment) | 25 |
| 16 | Lack of social capital to support cancer journey especially where patients must travel for care (relationships, support from family, friends, colleagues) | 17 |

20 operational radiotherapy units in order to respond adequately to its population demands.[37] Currently the only radiotherapy machines in the country are all located at the Uganda Cancer Institute (UCI) in the capital city Kampala. This has resulted in demand for services not being met (simply patients do not receive it), prolonged waiting times, compromises timing between the administration of radiation doses and eventually clinical and treatment outcomes. Even when potentially lifesaving radiotherapy treatment options are made available, there are challenges with the cost of treatment for a known cornerstone of curative therapy.

### The cost of treatment (surgery, radiotherapy, hormone therapy)
The cost of PC care is critical as there is limited access and availability to safe and reliable services including chemotherapy for PC patients in Uganda. The government of Uganda provides generic anticancer medicines for patients at the UCI at subsidised costs under the Universal Health Coverage (UHC) strategy to make available the WHO Essential anticancer Medicine List (EML).[38] For example, at the UCI, patients pay a subsidised fee of approximately $85 before accessing radiotherapy services.[6] In general, most patients with cancer in Uganda are not able to afford cancer therapy out of pocket and yet the UHC arrangement does not make available newer anticancer therapies. The proportion of anticancer agents especially the newer targeted therapies

on the 2019 WHO EML that are available on the 2016 Uganda National Essential Medicines List was about 70.5%. Essential medicines for PC including leuprolide, bicalutamide and abiraterone are often out of stock through the government UHC strategy.[39] Availability and prices of anticancer agents vary widely in the LMICs including Uganda. The low availability and unaffordability of anticancer agents often lead patients to turn to alternative care approaches.[40] PC is among the most common cancers treated with complimentary therapies in Uganda. The unavailability of critical medicines and curative treatments being out of reach have pushed many patients to seek alternative, traditional, Chinese and complementary medicines chiefly from potential anticancer medicinal plants.[6 41]

Overall, the study was able to collate and appraise the factors influencing diagnostic and treatment delays in PC care in Uganda. The study identified specific priority areas that are both high impact (significant cause of avoidable mortality) and modifiable (amenable to healthcare intervention) to direct resources and interventions to reduce disparities in access to PC care and improve outcomes. Future research investigating the interaction of the identified barriers and research priorities is necessary to build resilient and effective PC control programmes for sub-Saharan Africa. The findings contribute to the multi-country evaluation focusing on PC to enable strategic

priority setting and capacity building tailored for sub-Saharan Africa.

## Strengths and limitations

The Delphi technique was conducted by a multidisciplinary team following the CREDES guidelines.[16] The statements explored in the Delphi were identified through a systematic review. We were able to use both electronic web-based as well as face-to-face paper-based approaches to data collection. This study has some limitations: the response rates in round 1 and round 2 were relatively low (34% and 29%, respectively). However, the aim of this study was to have suitable representation to identify the top research priorities for PC care in Uganda rather than receiving a high response rate.

The discrepancy between the ratings of the statements across the two rounds as indicated above may have been introduced by two different panels with the differences in professional backgrounds of the participants participating in the two Delphi rounds. The reason for including a broader range of participants was due to the high attrition rate of panel 1 members. This is as a result of the fact that it is often impractical to secure the participation of all key knowledgeable individuals for all the Delphi rounds and consensus meeting. The study team agreed to invite more participants for round 2 by including cancer experts and clinicians working in settings beyond those invited in round 1 to include individuals from government hospitals, private-for-profit hospitals, private-not-for-profit hospitals, Ministry of Health and the university sector. Overall, we feel this breadth of participants involving cancer experts, clinicians, policy experts in and outside cancer management to evaluate the top research priorities for PC care in Uganda is a strength.

Further research is required to elicit patients' views. Although our questionnaires were sent to various groups and healthcare professionals from different clinical roles, most of the participants and experts in this study were oncologists and medical doctors. Future studies may need to consider other members of the healthcare team who were under-represented and may have alternative perspectives.

## CONCLUSION

This study has identified the top six research priorities for PC in Uganda. Our findings have implications for designing appropriate and contextual prostate care services locally to ensure men have fewer barriers to receiving earlier diagnosis and high-quality affordable treatment and survivorship care.

**Author affiliations**
[1]Social Aspects of Health Across the Lifecourse, MRC/UVRI and LSHTM Uganda Research Unit, Entebbe, Wakiso, Uganda
[2]Department of Global Health and Development, London School of Hygiene and Tropical Medicine, London, UK
[3]Radiation Oncology, MidCentral District Health Board, Palmerston North, New Zealand
[4]University of Zambia, Lusaka, Zambia
[5]Cancer Diseases Hospital, Lusaka, Zambia
[6]Department of Internal Medicine, Mulago Hospital/Makerere University, Kampala, Uganda
[7]Department of Medicine, Faculty of Medicine, Gulu University, Gulu, Uganda
[8]Health Services Research & Policy, London School of Hygiene and Tropical Medicine, London, UK
[9]King's College London, London, UK

**Acknowledgements** We thank the participants who participated in the Delphi technique and research team in Uganda where the study was conducted.

**Contributors** AA conceived the study idea and designed it with ASS, ADM and JS. AA, ASS, ADM, JS, DCL, MM and SM were engaged in the preparation and conduct of the Delphi study. ASS and RM collected the data. ASS, RM, DCL, MM, SM and AA led the writing of the paper and participated in the analysis. AA, ADM and JS supervised the overall study. All authors read and approved the final manuscript. ASS is the guarantor responsible for the overall content.

**Funding** The study was supported by funding from Wellcome's Institutional Strategic Support Fund grant no. 204928/Z/16/Z through the London School of Hygiene and Tropical Medicine.

**Competing interests** None declared.

**Patient and public involvement** Patients and/or the public were not involved in the design, or conduct, or reporting, or dissemination plans of this research.

**Patient consent for publication** Not applicable.

**Ethics approval** This study involves human participants and was approved by Uganda Virus Research Institute Research Ethics Committee (UVRI-REC) Ref: GC/127/21/09/859; Uganda National Council for Science and Technology (UNCST) Ref: HS1790ES; London School of Hygiene and Tropical Medicine (LSHTM) Ref: 26672. Participants gave informed consent to participate in the study before taking part.

**Provenance and peer review** Not commissioned; externally peer reviewed.

**Data availability statement** All data relevant to the study are included in the article or uploaded as supplementary information.

**ORCID iDs**
Andrew Sentoogo Ssemata http://orcid.org/0000-0003-0060-0842
Richard Muhumuza http://orcid.org/0000-0002-9931-7600
Janet Seeley http://orcid.org/0000-0002-0583-5272
Dorothy Chilambe Lombe http://orcid.org/0000-0002-5083-1801
Monde Mwamba http://orcid.org/0000-0003-3158-7724
Susan Msadabwe http://orcid.org/0000-0003-3887-3790
Amos Deogratius Mwaka http://orcid.org/0000-0001-7952-2327

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
