## [Reviewer comments · BMJ Open]

ARTICLE DETAILS

TITLE (PROVISIONAL)	Moving forward through consensus: a national Delphi approach to determine the top research priorities in prostate cancer in Uganda.
AUTHORS	Ssemata, Andrew; Muhumuza, Richard; Seeley, Janet; Lombe, Dorothy; Mwamba, Monde; Msadabwe, Susan; Mwaka, Amos; Aggarwal, Ajay

VERSION 1 – REVIEW

REVIEWER	Pedrosa Carrasco, Anna Philipps-Universitat Marburg, Faculty of Medicine, Research Group Medical Ethics
REVIEW RETURNED	18-Jul-2023

GENERAL COMMENTS	The study by Ssemata et al addresses an important issue by identifying key areas for research into prostate cancer in the Ugandan context. Specifically, the authors aim to identify the main barriers to seeking cancer care, accessing cancer care and receiving good quality cancer care in order to support the development of health system interventions to reduce inequalities and improve outcomes for patients receiving hospital care. For the most part, the manuscript summarises the study procedure well, but some methodological aspects are not discussed in detail and the implications of the study could be further explored. Abstract The abstract would benefit from revision. First, under Objectives it is not clear that the study aims to identify research priorities. Under Design, the word "qualitative" should be reconsidered as the study takes a quantitative approach. In the methodology section, the individual methodological steps could be presented in more detail. Introduction The introduction leads well into the rationale of the study. A somewhat more stringent thematic structure of paragraphs 3 and 4 could possibly add some value to this section of the manuscript. Methods Flexibility of the Delphi technique allows adaptation of the method to the requirements of the study. The authors present a modified Delphi technique for reaching consensus on research priorities, based on a systematic literature review. The consensus criteria were well defined. However, the manuscript lacks a rationale for the methodological decisions regarding the Delphi panel. The Delphi approach usually involves consulting one Delphi panel on a research topic over at least two rounds in order to reach a consensus within the panel. In the present study, two very diverse
--

	groups in terms of gender and occupation were recruited and surveyed with only three panelists participating in both rounds. Unfortunately, it is difficult to understand from the information given whether this was the result of high attrition or intended in the research protocol. Results The main results were clearly presented in tables. There is a notable discrepancy between the ratings of the statements across the two rounds, which is most likely due to the relevant differences in professional backgrounds, but which has not been addressed by the authors. Discussion The discussion provides an interesting overview of the identified research priorities, placed in the context of existing evidence. As the study aimed to identify research priorities, the discussion would have benefited from suggestions on how to use this information for future research projects. I would like to suggest discussing the potential bias introduced by two different panels participating in the two Delphi rounds.
--	--

REVIEWER	Pezzoli, Marta University of Florence
REVIEW RETURNED	16-Aug-2023

GENERAL COMMENTS	Q1: I believe this manuscript is well-written, detailed and the results describe well the challenges in diagnosis and treatment of prostate cancer in Uganda. Nevertheless, I have concerns about the participants at Delphi Consensus. Few invited clinicians effectively participated to consensus (34% and 29% at round 1 and 2 respectively), can you give a possible explanation of this low participation? I see that participants mostly changed between round 1 and 2 and the prevalence of their medical specialties also changed; for some topic this could have led to the different rankings shown in the results (for example, "lack of diagnostic services": ranking 50 in round 1 and ranking 92 in round 2). Could you explain the reason for this choice and how this could have influenced the final results?
--

VERSION 1 – AUTHOR RESPONSE

Reviewer 1

Dr Anna Pedrosa Carrasco

The study by Ssemata et al addresses an important issue by identifying key areas for research into prostate cancer in the Ugandan context. Specifically, the authors aim to identify the main barriers to seeking cancer care, accessing cancer care, and receiving good quality cancer care in order to support the development of health system interventions to reduce inequalities and improve outcomes for patients receiving hospital care. For the most part, the manuscript summarises the study procedure well, but some methodological aspects are not discussed in detail and the implications of the study could be further explored.

Abstract

The abstract would benefit from revision. First, under Objectives it is not clear that the study aims to identify research priorities.

Thank you for the observation. We have modified the objective to make it clear. It now reads as follows:

To identify key areas for research in prostate cancer (PC) in the Ugandan context by establishing the major health system, socioeconomic and clinical barriers to seeking, reaching, and receiving high quality cancer care. Page 2; paragraph 1

Under Design, the word "qualitative" should be reconsidered as the study takes a quantitative approach.

Thank you for the observation. We have deleted the word qualitative from the design section and replaced it with modified. Page 2.

In the methodology section, the individual methodological steps could be presented in more detail.

Thank you for the observation. We have added more detail to the methods section. Page 2.

Introduction

The introduction leads well into the rationale of the study. A somewhat more stringent thematic structure of paragraphs 3 and 4 could possibly add some value to this section of the manuscript.

Thank you for your comments. We have merged and revised paragraphs 3 and 4 to make a succinct and clear lead into the rationale. Please see page 4, paragraph 3.

Methods

Flexibility of the Delphi technique allows adaptation of the method to the requirements of the study. The authors present a modified Delphi technique for reaching consensus on research priorities, based on a systematic literature review. The consensus criteria were well defined. However, the manuscript lacks a rationale for the methodological decisions regarding the Delphi panel.

Thank you for this comment. This feeds in with our response below. The intention was to repeat the Delphi process within the same panel of participants but due to high attrition rate this was extended to include participants outside of the major specialist cancer centres to involve those working in district general hospitals and primary care. Please see page 6, paragraph 3 of the data collection section

The Delphi approach usually involves consulting one Delphi panel on a research topic over at least two rounds in order to reach a consensus within the panel. In the present study, two very diverse groups in terms of gender and occupation were recruited and surveyed with only three panellists participating in both rounds. Unfortunately, it is difficult to understand from the information given whether this was the result of high attrition or intended in the research protocol.

Thank you, whilst we did intend to survey the same panellists again from round 1, we experienced a high attrition of the participants involved in round 1 for round 2 either due to non-completion of the forms or incomplete forms despite the follow-up with a reminder phone call within a four-week period and face-to-face meetings with the participants to ensure completion of the questionnaire. The study team agreed to invite more participants for round 2 by expanding the geographical scope of the study to include cancer experts and clinicians working in settings beyond those invited in round 1 from the Uganda Cancer Institute, to include individuals from government hospitals, Private-for-profit hospitals, Private-not-for-profit hospitals, Ministry of Health, and the university sector.

Overall, we feel this breadth of participants involving g cancer experts, clinicians, policy experts in and outside cancer management to evaluate the top research priorities for prostate cancer care in Uganda is a strength. We have included a statement in the methods section of the manuscript to explain this attrition and between Round 1 and 2 and the rationale for inviting participants outside of the specialist centres. Please see page 6, paragraph 3 of the data collection section.

Results

The main results were clearly presented in tables. There is a notable discrepancy between the ratings of the statements across the two rounds, which is most likely due to the relevant differences in professional backgrounds, but which has not been addressed by the authors.

Thank you for the comment. We have addressed this in the discussion section in line with the comment regarding the potential bias resulting in the change in participants. Please see Page 12, paragraph 2.

Discussion

The discussion provides an interesting overview of the identified research priorities, placed in the context of existing evidence. As the study aimed to identify research priorities, the discussion would have benefited from suggestions on how to use this information for future research projects.

Thank you for the observation. We had added a paragraph on how this information can be used for future research. Please see page 14, after the cost of treatment section.

I would like to suggest discussing the potential bias introduced by two different panels participating in the two Delphi rounds.

There could be potential bias introduced by two different panels participating in the two Delphi rounds as it is often impractical to secure the participation of all key knowledgeable individuals for all the Delphi rounds and consensus meeting. However, we were able to include participants from different health service strata (primary and secondary care), professional backgrounds and specialities to provided insightful perspectives towards prioritising research needs for prostate cancer in the Ugandan context. The priorities chosen reflect important challenges including cancer awareness, increasing diagnostic and treatment capacity as well as addressing the high costs of care, which are not likely to have substantially changed with additional Delphi rounds or participants. We have added the potential limitation the different profile of participants in the two rounds. Please see Page 12, paragraph 2 and Page 15, paragraph 2 under strengths and limitations section.

Reviewer: 2

Dr. Marta Pezzoli, University of Florence

Comments to the Author:

Q1: I believe this manuscript is well-written, detailed and the results describe well the challenges in diagnosis and treatment of prostate cancer in Uganda. Nevertheless, I have concerns about the participants at Delphi Consensus. Few invited clinicians effectively participated to consensus (34% and 29% at round 1 and 2 respectively), can you give a possible explanation of this low participation?

Thank you. As per our response to review 1,

Whilst we did intend to survey the same panellists again from round 1, we experienced a high attrition of the participants involved in round 1 for round 2 either due to non-completion of the forms or incomplete forms despite the follow-up with a reminder phone call within a four-week period and face-to-face meetings with the participants to ensure completion of the questionnaire.

The study team agreed to invite more participants for round 2 by expanding the geographical scope of the study to include cancer experts and clinicians working in settings beyond those invited in round 1 from the Uganda Cancer Institute, to include individuals from government hospitals, Private-for-profit hospitals, Private-not-for-profit hospitals, Ministry of Health, and the university sector.

Overall, we feel this breadth of participants involving cancer experts, clinicians, policy experts in and outside cancer management to evaluate the top research priorities for prostate cancer care in Uganda is a strength. We have included a statement in the methods section of the manuscript to explain this attrition and between Round 1 and 2 and the rationale for inviting participants outside of the specialist centres. Please see Page 6, paragraph 3. We also acknowledge the low response as a potential limitation of this study in the strengths and limitations section of the manuscript. Please see Page 12, paragraph 2 and Page 15, paragraph 2 under strengths and limitations section.

I see that participants mostly changed between round 1 and 2 and the prevalence of their medical specialties also changed; for some topic this could have led to the different rankings shown in the results (for example, "lack of diagnostic services": ranking 50 in round 1 and ranking 92 in round 2). Could you explain the reason for this choice and how this could have influenced the final results?

Participants mostly changed between round 1 and 2 due to low response rates from participants invited during round 1. The study team agreed to invite more participants for round 2 by expanding the geographical scope of the study to include cancer experts and clinicians working in settings beyond those invited in round 1 from the Uganda Cancer Institute, to include individuals from government hospitals, Private-for-profit hospitals, Private-not-for-profit hospitals, Ministry of Health, and the university sector.

This could have influenced the results of this study, particularly the importance of early diagnosis given we included more participants from district general hospitals and primary care. However, it is also important to note is that the study team invited additional participants for round 2 from centres that provide cancer care services and work in liaison with those centres in round 1. Overall, we believe the diversity of specialties remains critical in providing a comprehensive evaluation of the top research priorities for prostate cancer care important in the development of a holistic national prostate cancer research program in Uganda.

Once again, thank you very much for reviewing our manuscript and considering it for publication with BMJ Open. We look forward to hearing from you.

VERSION 2 – REVIEW

REVIEWER	Pedrosa Carrasco, Anna Philipps-Universitat Marburg, Faculty of Medicine, Research Group Medical Ethics
REVIEW RETURNED	09-Nov-2023

GENERAL COMMENTS	The authors have acknowledged my concerns about the two panels responding to each round and have discussed these in their manuscript. I still consider this to be an important methodological flaw in this study, but I recommend publication of the results due to the relevance of the topic.
---